# The GlycoPaSER Prototype as a Real-Time N-Glycopeptide Identification Tool Based on the PaSER Parallel Computing Platform

**DOI:** 10.3390/ijms24097869

**Published:** 2023-04-26

**Authors:** Gad Armony, Sven Brehmer, Tharan Srikumar, Lennard Pfennig, Fokje Zijlstra, Dennis Trede, Gary Kruppa, Dirk J. Lefeber, Alain J. van Gool, Hans J. C. T. Wessels

**Affiliations:** 1Translational Metabolic Laboratory, Department of Laboratory Medicine, Radboud Institute for Molecular Life Sciences, Radboud University Medical Center, 6525 GA Nijmegen, The Netherlands; gad.armony@radboudumc.nl (G.A.); fokje.zijlstra@radboudumc.nl (F.Z.); dirk.lefeber@radboudumc.nl (D.J.L.); alain.vangool@radboudumc.nl (A.J.v.G.); 2Bruker Daltonics GmbH & Co. KG, 28359 Bremen, Germany; sven.brehmer@bruker.com (S.B.); tharan.srikumar@bruker.com (T.S.); lennard.pfennig@bruker.com (L.P.); dennis.trede@bruker.com (D.T.); gary.kruppa@bruker.com (G.K.); 3Department of Neurology, Donders Institute for Brain, Cognition and Behavior, Radboud University Medical Center, 6525 GA Nijmegen, The Netherlands

**Keywords:** glycoproteomics, real-time search, results-dependent acquisition (RDA), PaSER, GlycoPaSER

## Abstract

Real-time database searching allows for simpler and automated proteomics workflows as it eliminates technical bottlenecks in high-throughput experiments. Most importantly, it enables results-dependent acquisition (RDA), where search results can be used to guide data acquisition during acquisition. This is especially beneficial for glycoproteomics since the wide range of physicochemical properties of glycopeptides lead to a wide range of optimal acquisition parameters. We established here the GlycoPaSER prototype by extending the Parallel Search Engine in Real-time (PaSER) functionality for real-time glycopeptide identification from fragmentation spectra. Glycopeptide fragmentation spectra were decomposed into peptide and glycan moiety spectra using common N-glycan fragments. Each moiety was subsequently identified by a specialized algorithm running in real-time. GlycoPaSER can keep up with the rate of data acquisition for real-time analysis with similar performance to other glycoproteomics software and produces results that are in line with the literature reference data. The GlycoPaSER prototype presented here provides the first proof-of-concept for real-time glycopeptide identification that unlocks the future development of RDA technology to transcend data acquisition.

## 1. Introduction

Mass-spectrometry-based proteomics has become a staple method when studying proteins in complex mixtures [1,2,3]. The most notable approach is bottom-up proteomics using LC-MS/MS in which proteins are digested into peptides which are then separated using liquid chromatography (LC), ionized, and measured using tandem mass spectrometry (MS/MS). Acquired fragmentation spectra are then post hoc searched against a protein sequence database to identify peptide sequences (with modifications) and infer protein identifications. The recent introduction of the Parallel Search Engine in Real-time (PaSER [4]) enabled routine real-time protein database searching using peptide fragmentation spectra during sample measurement with the timsTOF instrument using Parallel Accumulation SErial Fragmentation in Data Dependent Acquisition mode (dda-PASEF [5]). Real-time data processing not only solves common computational bottlenecks and data stewardship challenges in typical proteomics workflows, but also opens up unique opportunities to optimize data acquisition on-the-fly. The potential benefits of this concept were demonstrated on other platforms [6,7,8,9,10] where MS/MS precursor selections were modified according to real-time analysis results, going deeper with identification and quantification. To this end, PaSER can communicate with an Application Programming Interface (API) on the acquisition computer to guide PASEF data acquisition and schedule precursor ions for reanalysis using individually optimized parameters when needed. The mass spectrometer receives direct feedback based on the results that it is producing; these new kinds of data are available for the acquisition logic which opens a whole new field of research in mass spectrometry with unprecedented possibilities to enhance experimental outcomes.

The potential of real-time results-dependent acquisition (RDA) is of particular interest for the analysis of glycosylated peptides in complex mixtures within the field of glycoproteomics. Protein glycosylation is a key modulator of protein biology that has been shown to dynamically change in various genetic or acquired diseases [11,12]. Glycoproteomics enables proteome-wide characterization of protein glycosylation at the level of individual glycosylation sites, which provides unique possibilities for biomarker applications and understanding of the intricate biology underlying this complex modification class. Characterization of glycopeptides by LC-MS/MS is inherently challenging because of the relatively low intensities of the glycopeptide precursors and their fragmentation behaviour in collision-induced dissociation experiments. The diverse fragmentation behaviour is due to intrinsic physicochemical differences between the peptide and glycan moiety of these hybrid amino acid–sugar copolymers. Even more so, the glycoproteome contains an overwhelming variation in combinations of peptide sequences and glycan structures [13]. This complicates MS/MS data acquisition since optimal activation energies to achieve rich fragmentation spectra are harder to predict from the *m/z* or collisional cross section of precursor ions. Here, the use of real-time glycopeptide identification results together with fragmentation spectrum information to guide glycopeptide data acquisition offers an enticing possibility to advance glycoproteomics.

Applying the concept of using on-the-fly results for the adjustment of acquisition parameters in glycoproteomics requires glycopeptide identification capabilities that are currently unavailable on the PaSER platform. Moreover, the great diversity in glycan structures that can occupy a single glycosylation site in proteins are beyond the limits of regular variable modification in most proteomics software. Hence, specialized algorithms are required to determine in real-time the composition and/or structure of glycan moieties. To enable such real-time glycopeptide searches on PaSER, we set out to develop GlycoPaSER which takes advantage of the available real-time protein database search engine “ProLuCID” [14]. Our strategy is to decompose the original hybrid glycopeptide fragmentation spectrum into two composite spectra that contain either peptide fragmentation products or glycan fragmentation products (Appendix A).

In this work, we share proof-of-concept for real-time glycopeptide identification from online dda-PASEF measurements using the newly developed GlycoPaSER prototype software. We evaluated its glycopeptide identification performance by comparing plasma glycoproteomics results to offline MSFragger-Glyco [15] output and available reference data from the literature. In addition, we assess its computational performance in relation to the instrument PASEF duty cycle and investigate the potential gain of using optimized collision energies for future applications.

## 2. Results

### 2.1. GlycoPaSER Prototype Design

N-glycopeptide fragmentation spectra contain a mixture of peptide, glycan, and peptide + glycan fragment ions from which both the peptide and glycan moiety need to be elucidated. Our strategy is to decompose hybrid glycopeptide fragmentation spectra into separated peptide and glycan moiety spectra. This would enable the characterization of each moiety separately by using specialized algorithms and avoid the incorrect assignment of peptide fragments as glycan fragments and vice versa. We aimed to achieve real-time spectrum decomposition in three consecutive steps by developing a decomposer module for PaSER. Figure 1A depicts the data flow in the decomposer module of GlycoPaSER with the following three major steps:Filter for glycopeptide fragmentation spectra by the use of oxonium ion signatures.For each selected glycopeptide spectrum, identify the peptide + HexNAc mass by searching for the N-glycan core fragmentation pattern.Generate respective peptide and glycan moiety fragmentation spectra using the peptide + HexNAc mass and removing glycan fragment peaks after charge deconvolution.

Step 1: Glycopeptide fragmentation spectra are selected for subsequent spectrum decomposition by the presence of oxonium ions which are glycopeptide diagnostic fragments. Oxonium ions are characteristic fragments of the glycan moiety (Figure 1B and Appendix A). If their predefined masses are detected at sufficient intensity, the decomposer module will send the spectrum to the glycan core pattern finder; otherwise, the spectrum is streamed to ProLuCID to be searched as a regular non-glycopeptide.

Step 2: Upon collisional activation of a glycopeptide precursor ion with the appropriate activation energy, the glycan moiety is fragmented at glycosidic bonds. This results in a fragment series with mass differences corresponding to the sequence of the sugars along the glycan, including the common N-glycan core sequence of Asn-HexNAc-HexNAc-Hex-Hex-Hex. The first fragment in this Y-ion series is the deglycosylated peptide moiety (or Y_0_ ion); therefore, if we find the fragment ions which follow this pattern, we can deduce the mass of the peptide moiety and pass the spectrum to the modifier submodule. Figure 1B shows an illustration of a glycopeptide fragmentation spectrum with the glycan core fragmentation pattern highlighted.

Step 3: The spectrum modifier generates peptide and glycan moiety composite spectra with appropriate virtual precursor ion masses for each moiety. The peptide moiety spectrum is generated by modifying the original precursor mass to [M + HexNAc]^1+^, charge deconvoluting the spectrum, removing oxonium ion peaks, and removing all Y-ions by removing all peaks with a mass larger than the modified precursor mass (Appendix A). The glycan moiety spectrum is generated by charge deconvolution and adding the calculated glycan moiety mass to the spectrum metadata. The last step of the decomposer is streaming the modified spectra to their respective identification modules, the peptide moiety spectrum to ProLuCID, and the glycan moiety spectrum to a glycan composition generator.

For peptide moiety elucidation, we made use of the existing ProLuCID search engine “as is” without any optimization of this algorithm. To perform basic glycan identification, we developed a PaSER module that generates all possible glycan compositions by exhaustively checking all combinations of sugar building blocks for compositions that fit the determined glycan moiety mass within user-defined constraints (Figure 1C). Each GlycoPaSER step for MS/MS spectrum decomposition will be explained in detail in the subsequent sections.

### 2.2. Oxonium Ion Filter-Based Selection of Glycopeptide MS/MS Spectra

The decomposer module has several parameters that need to be set for it to perform well. We determined these parameters in a data-driven manner using data of 10 plasma samples from healthy controls. We used MSFragger-Glyco [15] to search these data and used the search results as a reference for parameter setting, testing, and benchmarking.

The goal of the first step in the decomposer module, the oxonium filter, is to filter out as many non-glycopeptide spectra while retaining as many glycopeptide spectra as possible. To achieve this, we determined two parameters: which oxonium ions to use and how intense they should be (Figure 2A,B). We checked 42 oxonium ions (Appendix A) and found that 99.8% of the glycopeptide fragmentation spectra (as determined by using MSFragger) contained a HexNAc-Hex ion (*m/z* 366.1395). The glycopeptide spectra that did not contain this ion (0.2%) could be accounted for by one of five other ions (HexNAc, Neu5Ac, Neu5Ac-H_2_O, and HexNAc-Hex-Neu5Ac, Figure 2A). However, some non-glycopeptide spectra also contained a mass matching one of these six oxonium ions (Figure 2A). These spectra can be filtered out by requiring the presence of more than one ion per spectrum while considering more ions (Appendix A), but an even better classifier is provided by the summed relative intensity from all the detected oxonium ions (Appendix A). We selected a threshold for the relative intensity sum such that only 5% of the non-glycopeptide spectra passed the filter (false positives) while retaining 99.1% of the glycopeptide spectra (true positives) as shown in Figure 2B.

### 2.3. N-Glycan Core Pattern Finder-Based Spectrum Decomposition into Composite Peptide and Glycan Moiety Fragmentation Spectra

The glycopeptide spectra that pass the oxonium ion filter are passed on to the pattern finder, which finds the best match for the N-glycan core pattern in each spectrum. First, we set which pattern to search for; the N-glycan core has five monosaccharides, which yields a collision-induced dissociation (CID) fragmentation pattern of six peaks: p, p + HexNAc, p + 2 HexNAc, p + 2HexNAc + Hex, p + 2HexNAc + 2 Hex, p + 2HexNAc + 3 Hex, where p is the mass of the peptide moiety. In addition, two N-glycan core fragment peaks have been reported to be commonly generated in CID experiments [16,17], one originating from deamidation of the glycosylated asparagine, and the other from cross-ring fragmentation of the proximal HexNAc residue. Therefore, the pattern we use is composed of these eight peaks (Figure 1B). When searching for a pattern, we measure the mass distances (the mass offset between the pattern peaks) from the [M + HexNAc]^1+^ (Y_1_ ions) as the reference peak since it will later be used as pseudo precursor mass when generating the composite peptide moiety spectrum.

We used the plasma data and MSFragger results to evaluate several parameters for the fragmentation pattern finder. We checked which parameter and values exclude wrong patterns. The minimum mass and intensity of the reference peak were the most relevant. A minimum mass for the reference peak excludes any patterns matched with small fragment ions below the peptide moiety mass. Theoretically, the lightest tryptic glycopeptide is GGGNK.S with a mass of 431 Da; however, in practice, the lightest glycopeptide we identified was ANISHK with a mass of 688 Da or 891 Da with the proximal HexNAc. We also set a minimum intensity for the reference peak since it is usually one of the most intense peaks of the fragmentation spectrum (Appendix A), and considering all peaks, including noise signals, would be too computationally intensive for running in real-time. We therefore considered fragment ions with a base peak intensity greater than 10% and with a mass greater than 850 Da as reference peaks.

Searching for the pattern in a spectrum usually yields multiple pattern matches (multiple sets of peaks that match the pattern), especially when we allow for partial pattern matches. To select the best pattern match, we ranked them by sorting by the number of peaks matched to the pattern and then by intensity, such that rank one would have the most peak matches and with the highest intensity. We then determined if the pattern that was ranked one was indeed the correct pattern match in two ways. In the first way, we used MSFragger identification results as a reference to label the correct pattern match when its respective peptide moiety mass corresponded with MSFragger results within 0.02 Da mass error tolerance. In the second way of evaluating the ranking, we generated 10 modified peptide moiety spectra from each fragmentation spectrum using the peptide moiety mass of the top 10 ranked patterns. These spectra, along with the original unmodified spectra, were submitted to protein sequence database searches by ProLuCID. For each spectrum, a pattern match was labelled as correct if its corresponding modified spectrum had the best peptide spectrum match (PSM). If the unmodified spectrum yielded the best PSM, all the patterns were labelled as incorrect.

Using the correct pattern labelled according to ProLuCID, we determined the last parameter for the pattern finder, the minimum number of ions matching the pattern. As we can see in Figure 2C, about half of the correct patterns had eight matches, matching the entire pattern. Nonetheless, there were still correct patterns with only two matches (3.6%). Therefore, we chose to consider all pattern matches irrespective of the number of peaks that support it and relied on our pattern ranking to select the correct pattern. Next, we evaluated how accurate our ranking method was in more detail by analyzing the distributions of ProLuCID and MSFragger results over different classes. In most of the spectra (70% and 76%, respectively) the top-ranking pattern was indeed the correct pattern (Figure 2D). In 25% and 13% of the cases, the correct pattern match was ranked lower so that the decomposer produced the wrong peptide moiety composite spectrum. In many of these cases, the top-ranking (but wrong) pattern was ranked higher than the correct pattern since it had an extra peak match, but the correct pattern had a higher intensity. In a small percentage (5% and 13%, respectively), none of the patterns were correct, either because none of them matched the identified peptide moiety (red), or because the spectrum was not of a glycopeptide (purple) and therefore there was no pattern to be found. Based on this analysis, we concluded that the accuracy of this ranking method is sufficient to be used for selecting the correct pattern match for glycopeptide fragmentation spectra in the GlycoPaSER prototype.

### 2.4. GlycoPaSER Real-Time Computational Performance

Prior to the real-life testing of GlycoPaSER during timsTOF Pro measurements, we verified that it could keep up with the rate at which fragmentation spectra are generated. The acquisition duty cycle of a typical timsTOF method is depicted in the center of Figure 3, starting with an MS frame which is used to decide what precursors to fragment, followed by PASEF MS/MS frames (tims separation ramp), where in each frame, several of the selected precursors are fragmented for identification [5]. Across all 10 plasma samples, the average time to acquire a precursor fragmentation spectrum was 107 ms, which means that GlycoPaSER must fully process spectra at >9 Hz to be able to run in real-time. PaSER runs in parallel to the acquisition duty cycle, and when acquisition of its last MS/MS frame is finished, it is streamed from the acquisition computer to the PaSER box where spectra are processed and searched (Figure 1A and Figure 3). For testing, data files were streamed to PaSER with an acquisition simulator which showed that GlycoPaSER was able to process and search all the fragmentation spectra when they were sent every 35 ms (~30 Hz acquisition rate), which is easily compatible with the PASEF data acquisition method used in this work. We further investigated whether the GlycoPaSER modules we introduced would bottleneck real-time data processing by timing each individual component. We conclude that, based on our results in Figure 3 and Appendix A, it appears that GlycoPaSER modules are not rate-limiting with respect to current search parameters.

### 2.5. GlycoPaSER Real-Time Peptide moiety Identification Performance

For benchmarking the glycopeptide identification performance, we compared the results from GlycoPaSER to the results from MSFragger for the same 10 plasma control samples. We first compared the sequence motif for N-glycosylation–NXS/T, where X is any amino acid but not Proline. The distribution between both variants was very similar (Figure 4A). The distribution was also highly similar to the distribution of the glycosylation sites annotated in Uniprot. Moreover, according to the results from PaSER, 76% of the peptide moiety sequences contained the N-glycosylation sequon which increases confidence in these identifications, since unlike MSFragger, GlycoPaSER does not yet filter out peptides that lack the N-glycosylation sequon.

We next compared glycopeptide identifications at three levels: PSM, peptide moiety sequences, and glycoproteins. Comparing the glyco-PSMs revealed that about a third of the spectra identified by either tool were not identified by the other one (Figure 4B, red and blue). This was expected to some extent since the two identification tools follow different approaches to glycopeptide identification. GlycoPaSER uses glycopeptide decomposition while MSFragger uses an open mass search [18]. On the other hand, this difference in approaches strengthens confidence in overlapping results, where the two different approaches lead to the same identification. Indeed, when both tools identified the same spectrum, it was the same identification in most cases (93%) (Figure 4B, green). Investigating the 14,069 spectra uniquely identified by MSFragger revealed that in ~75% of the cases, PaSER did not produce an identification but was close. In ~50% (Figure 4B *), the glycopeptide decomposer module found the correct peptide moiety mass, but the modified peptide moiety spectrum was not identified by ProLuCID. In the other ~25% (Figure 4B **), the glycopeptide decomposer module found the correct core fragmentation pattern, but it was not selected since it did not have the highest rank. These observations indicate that improvements to ProLuCID and the glycopeptide decomposer would further increase the glycopeptide identification performance. The qualitative comparison between both software at peptide moiety and glycoprotein levels show excellent agreement based on the large overlap in consistently detected sequences and glycoproteins for at least eight out of the ten control samples (Figure 4C).

### 2.6. Glycan Composition Generation and Glycoproteome Coverage

We developed a database-independent approach to the glycan moiety identification which allowed us to identify unexpected glycans that are not listed in the database. This is especially useful when analyzing samples from patients where disease-specific glycans can be observed. The current glycan moiety identification is simple, using only the glycan moiety mass to generate possible compositions even though glycan fragments hold much more information. Nonetheless, it generates valuable information since for most glycopeptides, we could find only one or two possible compositions (Appendix A). Even though the current GlycoPaSER prototype does not yet use glycan fragments in the glycan composition generation, for 78% of the glycoPSMs, both PaSER and MSFragger generated the same glycan composition.

To assess the glycoproteome coverage of the GlycoPaSER output from the 10 human control samples, we visualized 123 identified N-glycosylation sites together with 70 unique glycan moiety mass offsets in a chord diagram (Figure 5A, Appendix A) that reflects the intricate complexity of the glycoproteome. On the top, we can see the identified N-glycome, where we can observe that the most frequently detected glycan masses correspond with complex di- and tri-antennary glycans which are the dominant glycans of the plasma N-glycome [19,20]. On the bottom of the plot, we can appreciate the microheterogeneity of the glycosylation sites. In addition, many resident plasma proteins were detected with glycan masses corresponding with known glycans from the literature such as tri-antennary complex glycans at α1-acid-glycoprotein, ceruloplasmin, and α-2-HS-glycoprotein or fucosylated truncated complex glycans at immunoglobulin heavy constant gamma proteins or high mannose glycans at complement component proteins C3 and C4b [19]. On average, we identified 2.1 N-glycosites per protein from resident plasma proteins that span the top six orders of magnitude in abundance (Figure 5B). These results are a significant improvement over our previous characterization of the baseline plasma glycoproteome where the same samples were analyzed using conventional Qq-TOF instrumentation in combination with ProteinScape and Mascot software [21]. The glycan moiety masses for the vast majority of N-glycosites correspond with glycan compositions that are listed in the GlyGen reference database, as shown in Figure 5C for the three illustrative glycoproteins examples of complement component C4b, immunoglobulin µ, and serotransferrin. Combined, these results show that the GlycoPaSER output for the plasma glycopeptide samples correlate well to the available reference data at high sensitivity.

### 2.7. Potential of On-the-Fly Acquisition Parameters Adjustment for Improved MS/MS Data Acquisition

To demonstrate how using the real-time glycopeptide identification results can improve the data quality, we analyzed a set of measurements where the same sample was measured at different collision energy settings. The collision energy (CE) in the timsTOF is determined by the measured mobility value (with a linear scale); however, we found glycopeptides with similar mobility values but different optimal collision energy values (Figure 6A). This indicated that modifying the CE setting may improve the identification quality of glycopeptides. Therefore, we selected the measurement with the optimized default CE setting as a reference and the glycopeptide identification results (with other CE settings) were all matched to the reference results. For each unique glycopeptide identification (unique peptide moiety sequence, glycan mass, and charge), the PSM with the highest score was selected. Figure 6B shows the collision energy and mobility values for all these best-scoring PSMs. Interestingly, 61% of the glycopeptide identifications can be improved by using other collision energies (green dots).

To simulate how real-time modification of the acquisition parameters could generate higher-quality data, we generated a file with hybrid spectra by cherry picking for each precursor the spectrum with the optimal CE out of the seven CE settings (Figure 6B). We performed the glycopeptide search using PaSER for both the original file and the modified file and, as expected, when the spectra were replaced with spectra collected at a better CE, the PSM scores increased (Appendix A). Moreover, in this example the increase in PSM score was only reflected in spectra that passed the false discovery rate (FDR) control in the original file, while unidentified spectra, that can benefit the most from optimized CE, were not included. This demonstrates the unique potential of results-based on-the-fly adjustment of acquisition parameters in challenging glycoproteomics applications.

## 3. Discussion

The GlycoPaSER prototype described in this work is capable of real-time glycopeptide identification where it can keep up with data generation in a real-life plasma glycoproteomics experiment. Spectral decomposition of the glycopeptide spectrum into peptide and glycan moiety composite spectra enables the subsequent identification of each moiety with a highly significant overlap between our GlycoPaSER prototype and MSFragger-Glyco results. These results, combined with the observed correlation to available reference plasma glycoproteome data, corroborate the accuracy of glycopeptide identifications. We show for the first time the successful application of real-time search technology for glycoproteomics, significantly reducing data processing time, and with great potential for further development towards comprehensive glycoproteomics software with real-time acquisition optimization capabilities.

One of the strengths of the glycan core fragmentation pattern search is that it accepts all partial pattern matches, which should enable its application to disease-specific glycoforms with an abnormal N-glycan core sequence. The pattern matching performance may be further improved by, for example, machine learning models that include more pattern match characteristics for an even better performance. In addition, it can be expanded further for application to O-glycopeptide MS/MS spectra in order to develop GlycoPaSER into a generic glycoproteomics tool beyond N-glycosylation.

The ProLuCID database search engine that is embedded within GlycoPaSER can be further optimized for performance on peptide moiety fragmentation spectra. For example, glycopeptide fragmentation spectra often contain peptide fragment (b-, y-) ion series both with and without the proximal HexNAc, while at present, ProLuCID evaluates only the series with. Evaluating both ion series would not only increase peptide spectrum match confidence but would also lower the penalty a match receives for unexplained residual fragment ions to enhance the match scoring.

The glycan identification currently implemented in GlycoPaSER is simple, providing all possible (restrained) glycan compositions based on the glycan moiety mass alone. Yet, it is surprisingly effective, yielding only a few putative compositions per spectrum, which enables us to evaluate different strategies for the development of an MS/MS-driven database-independent glycan identification algorithm. Here, we plan on using information from glycan fragment ions to determine a minimal composition of the glycan as is often performed in manual spectrum annotation. For example, the presence of sialic-acid-containing oxonium ions excludes any possible composition lacking sialic acid. Our pursuit of a database-independent glycan identification approach is of particular interest for clinical applications, where in congenital disorders of glycosylation, for example, uncommon disease-specific glycoforms can be present that may not be listed in a database [20].

An attractive objective of real-time glycopeptide identification is to use the identification data to guide the acquisition, using the instrument in a smarter way to generate higher-quality data. We demonstrated here the potential for optimized collision energies to improve data quality. Other instrument parameters can also be evaluated, such as increasing the number of summed MS/MS scans for low signal-to-noise ratio fragmentation spectra. The software infrastructure for performing such on-the-fly acquisition guidance is already available through an instrument API and preliminary research is ongoing to determine which fragmentation spectra should be reacquired and how optimal parameters can be derived and used.

To our knowledge, this work documents the very first successful real-time glycoproteomics data processing for LC-MS/MS, which opens exciting avenues for the future development of RDA. This will facilitate the results-driven, on-the-fly optimization of acquisition parameters for higher-quality and deeper glycoproteomics data.

## 4. Materials and Methods

### 4.1. Sample Preparation

Plasma samples of 10 healthy human control subjects were received from the Sanquin blood bank (Nijmegen, The Netherlands) according to their protocols of informed consent. Samples were prepared as described in [21]. Briefly, 10 µL of plasma was denatured in 10 µL urea (8 M urea, 10 mM Tris-HCl pH 8.0) and reduced with 15 µL 10 mM dithiothreitol for 30 min at room temperature (RT). Reduced cysteines were alkylated through incubation with 15 µL 50 mM 2-chloroacetamide in the dark for 20 min at RT. Next, proteins were subjected to LysC digestion (1 µg LysC/50 µg protein) by incubating the sample at RT for 3 h. Then, samples were diluted with 3 volumes of 50 mM ammonium bicarbonate and trypsin was added (1 µg trypsin/50 µg protein) for overnight digestion at 37 °C. Glycopeptides were enriched using 100 µL Sepharose CL-4B beads slurry (Sigma, St. Louis, MO, USA) per sample well in a 0.20 µm pore size 96 multiwell filter plate (AcroPrep Advance, VWR, Radnor, PA, USA). The beads were washed three times with 20% ethanol and 83% acetonitrile (ACN), respectively, prior to sample application. The sample was then incubated on the beads for 20 min at room temperature on a shaking plate. The filter plate was then centrifuged, and beads were first washed three times with 83% ACN and then three times with 83% ACN with 0.1% trifluoroacetic acid (TFA). Next, glycopeptide eluates were collected by incubation of the beads with 50 µL milliQ water for 5 min at room temperature, followed by centrifugation.

### 4.2. MS Acquisition

Samples were measured using a nanoElute nanoflow liquid chromatograph (Bruker Daltonics) coupled online to a timsTOF Pro2 instrument (Bruker Daltonics) via a CaptiveSprayer nanoflow electrospray ionization source using acetonitrile as the nanoBooster dopant (Bruker Daltonics) [23]. Peptides were separated on an ELUTE FITEEN C18 reversed-phase column (0.075 mm ID × 150 mm length, 1.9 µm particles, 120 Å pore size, C18-AQ2 chemistry) operated at 45 °C using a linear increase of 5 to 43% acetonitrile in 0.1% formic acid and 0.02% trifluoroacetic acid over 25 min at a flow rate of 500 nL/min. Mass spectrometry measurements were performed in positive ionization mode using 0.2 bar N2 nanoBooster gas pressure and 1500 V capillary voltage as source conditions. Spectra were acquired within 0.7–1.5 1/K0 mobility and 50–4000 *m/z* ranges using 10 dda-PASEF ramps at 50.000 target intensity and 30 eV at 0.6 Vs/cm^2^ 1/K0 to 90 eV at 1.6 Vs/cm^2^ 1/K0 as the default collision energy. Collision energies were varied for selective experiments as follows: 20, 22, 24, 26, 28, 30, and 32 eV at 0.6 Vs/cm^2^ 1/K0 to 60, 66, 72, 78, 84, 90, and 96 eV at 1.6 Vs/cm^2^ 1/K0, etc.

### 4.3. Database Search Settings

PaSER database searches were conducted with version 2022c with the default parameters modified to match the glycoproteomics experiment. The database contained all human proteins which are labelled as secreted on Uniprot, downloaded on 22 November 2021. Peptide mass tolerance was set to 30 ppm with 3 isotopic peaks, the precursor mass range was set to 600–50,000 Da, and semi-tryptic enzyme digestion specificity was used. Variable modifications were set to oxidation of methionine, HexNAc on asparagine, and N-terminal ammonia loss. MS/MS spectra were considered to be deisotoped and decharged and the multistage activation mode was set to 1 (i.e., considering both neutral loss and non-neutral loss peaks). FDR was set to 1% at the protein level, noisy PSMs were filtered, and spectra display mode was set to 0 (i.e., including all PSMs for each sequence).

MSFragger searches were conducted with fragpipe 17.1, msfragger 3.4, and philosopher 4.1.1. The glyco-N-HCD parameters were adjusted to match the glycoproteomics experiment. Namely, the mass tolerance was set to 30 ppm with an isotope error of 0–3, the enzyme was set to trypsin (semi-specific), peptide length to 5–50, and *m/z* range to 600–20,000. Variable modifications included the following: oxidation of methionine and N-terminal ammonia loss. The glycan mass offsets were extracted for unique composition in the GlyGen glycan reference database [24]. The FDR was set to 1% at PSM, peptide, and protein levels. For glycan assignment and FDR, the GlyGen database downloaded on 22.4.2022 was filtered for unique compositions.

### 4.4. Parameters for the Glycopeptide Decomposer

The glycopeptide decomposer has several adjustable parameters. The minimum spectrum peak intensity was set to 15, the mass error for oxonium ions was set to 0.02 Da, and the minimal number of oxonium ions was set to 1 out of a list of six—366.139472 (HexHexNAc), 657.234889 (HexHexNAcNeuAc), 512.19793 (HexHexNAcdHex), 292.102693 (NeuAc), 274.092128 (NeuAc-H_2_O), and 204.0867 (HexNAc). The minimum relative oxonium ion intensity sum was set to 0.0047 (the ratio of the intensity of the oxonium ions to the total intensity of the fragmentation spectrum). The pattern to search for was set to be with offsets [−220.0821, −203.0794, −120.0423, 0, 203.0794, 365.1322, 527.185, 689.2378], where 0 offset is the reference mass, the peptide + HexNAc peak. The reference mass was set to be at least 850 Da with a base peak intensity of at least 0.1. The mass error for matching pattern peaks was set to 0.05 and pattern matches with at least 2 peaks were considered (the reference mass and at least one other peak). Multiple patterns found in a spectrum were ranked first by the number of pattern matches and then by the reference peak intensity, such that the top rank had the most pattern matches with the most intense reference mass.

### 4.5. Parameters for the Glycan Composition Generator

The glycan composition generator has several adjustable parameters. Building blocks were set to be Hex (162.05282), HexNAc (203.07937), dHex (146.05791), and Neu5Ac (291.09542). Each building block had a minimum and maximum set: [0, 12], [1, 7], [0, 3], and [0, 4], respectively. The mass error for matching a composition to the glycan moiety mass was set to 0.1 Da.

### 4.6. Determination of the Correct Pattern Using ProLuCID

To determine which of the multiple patterns found for each spectrum was correct, the top 10 ranking patterns were tested. Each spectrum was modified according to each of the patterns, resulting in up to 10 modified spectra per spectrum. All the spectra with modified precursor mass from each sample were written together to an ms2 file, and all the spectra with the original precursor mass were written to a second ms2 file. These files were uploaded and separately searched in PaSER; then, the results were merged and the PSMs originating from the same precursor were grouped. If the best-scoring PSM was from a modified spectrum, the corresponding pattern was labelled as correct, if the best-scoring PSM was from the spectrum with the original precursor mass, all the patterns were labelled as incorrect, and if none of the spectra yielded a PSM, that precursor was labelled as unknown and was not used in this analysis. Often, multiple patterns pointed to the same species, each pointing to a different isotope; if one of these patterns was labelled as correct, the other patterns were also labelled as correct since all of them led to the same identification.

### 4.7. Timing Glycopeptide Acquisition and Identification

We calculated the average acquisition time per precursor by averaging the value for each cycle. The value for each cycle was determined as the ratio between the cycle time to the number of precursors which were selected for acquisition in that cycle. The cycle time was the difference in the Time column between MS1 frames and MsMsType 0 in the Frames table of the analysis.tdf file. The number of precursors selected in a frame was given by how many precursors have that frame as their Parent in the precursors Table. 

The new glycopeptide identification components in PaSER (Appendix A) were timed for each spectrum analysis by logging the time during analysis. The ProLuCID identification timing (Figure 3) was more challenging due to the GPU parallelization; spectra were sent to ProLuCID faster than it could process and the times for identifying each 100-spectra batch were recorded and averaged.

### 4.8. PaSER Identification Performance

The glycosylation motif fractions from Uniprot (Figure 4A) were calculated for all reported glycosylation sites for all the proteins present in the database used for the PaSER database searches.

When comparing identifications between PaSER and MSFragger, we considered the identification to be the same if the peptide moiety sequence was identical and the glycan moiety mass was within a mass error of 0.05 and 3 isotope peaks (for when the non-monoisotopic precursor mass was selected for fragmentation).

## Figures and Tables

**Figure 1 ijms-24-07869-f001:**
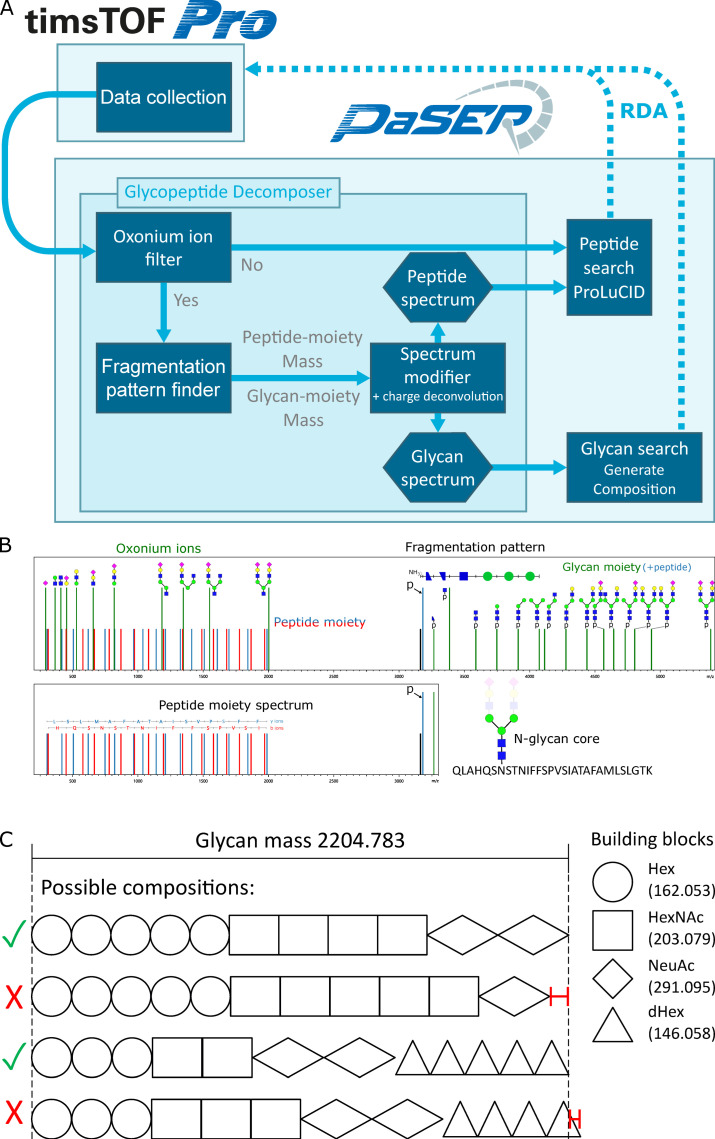
Glycopeptide identification strategy. (**A**) Data flow scheme for real-time glycopeptide identification in PaSER. The broken arrows indicate search-results-dependant acquisition (RDA) which is not yet implemented. (**B**) A schematic illustration of a glycopeptide fragmentation spectrum annotated with the features used for decomposition and identification (p is peptide moiety mass (Y_0_ ion)). (**C**) Schematic example for how glycan moiety compositions are generated in PaSER using the glycan moiety mass of the spectrum in (**B**). ✓ and ✕ indicate compositions that fit or do not fit the glycan mass, respectively. The red whiskered line on the right represent the mass deviation of the incorrect compositions from the glycan mass.

**Figure 2 ijms-24-07869-f002:**
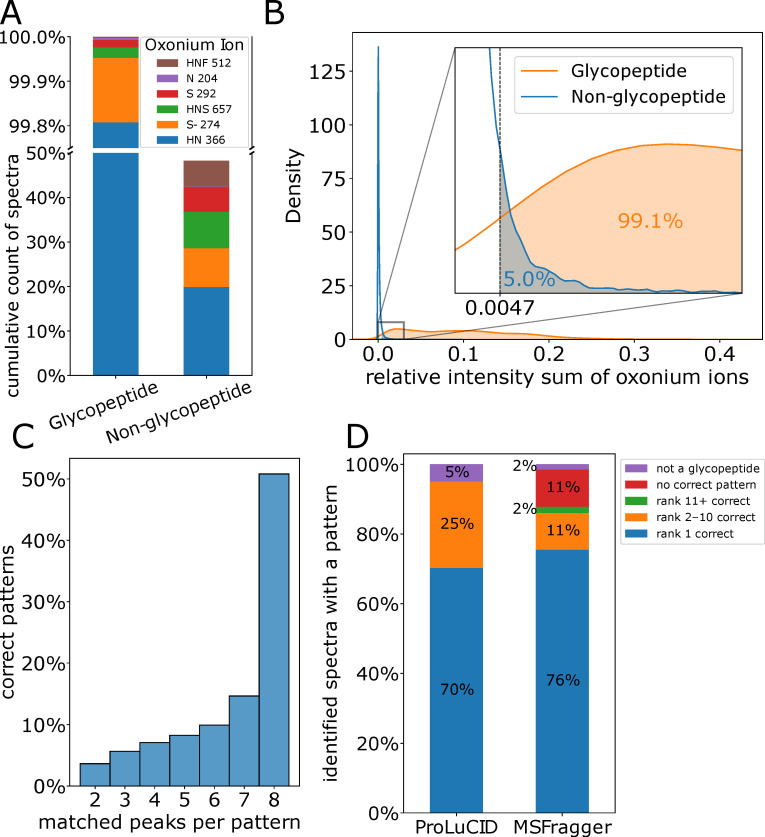
Decomposer parameters optimization. (**A**) Cumulative count of spectra that contain a mass corresponding to the oxonium ions. (**B**) Distribution of the oxonium ions relative intensity sum and the threshold that was picked. (**C**) Distribution of the number of matching ions in the correct N-glycan core fragmentation pattern (patterns which lead to a peptide moiety identification). (**D**) Performance of the fragmentation pattern ranking method, which rank does the correct pattern has according to the identification of ProLuCID or MSFragger.

**Figure 3 ijms-24-07869-f003:**
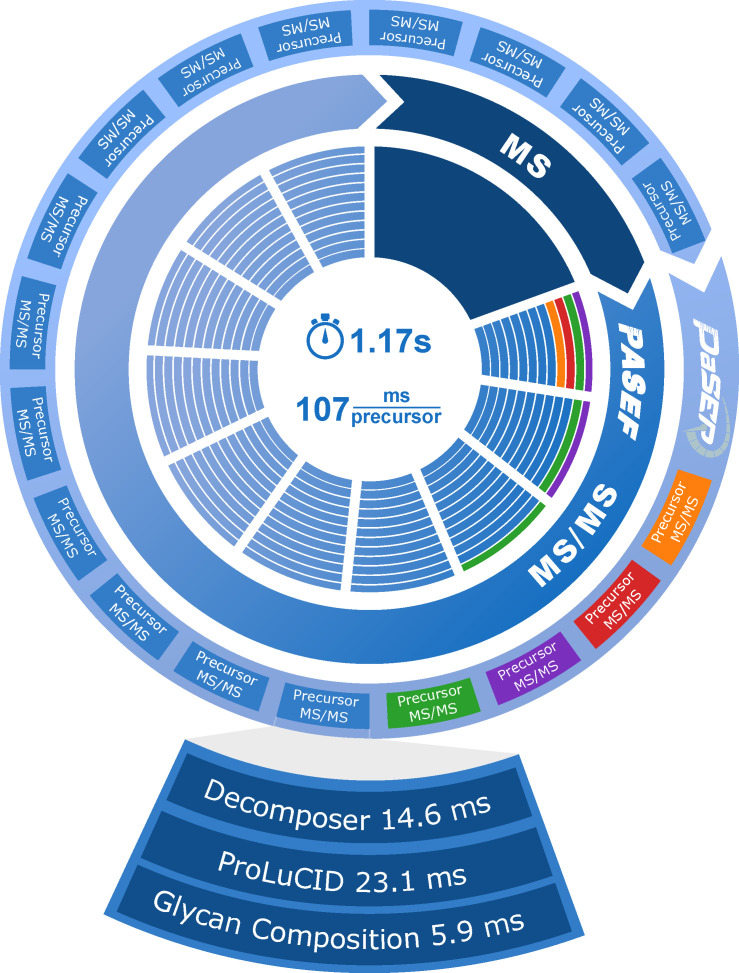
Computational performance of glycoPaSER. The PASEF acquisition cycle is depicted in the center where multiple precursors are acquired in parallel. The PaSER identification is depicted on the outer circle where the colored precursors are an example depicting the parallel but asynchronous nature of GlycoPaSER. The average timings for precursor data acquisition are given in the center and the average timings of glycopeptide identification are indicated in the enlarged precursor MS/MS.

**Figure 4 ijms-24-07869-f004:**
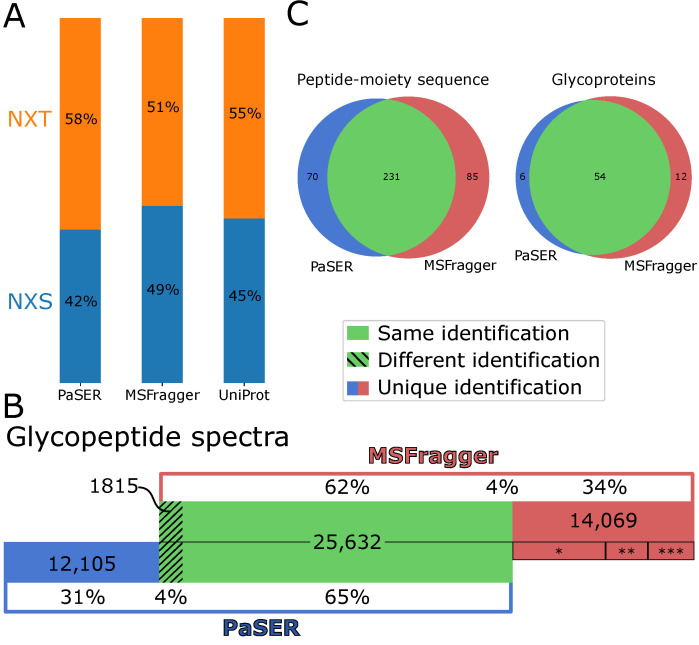
Glycopeptide identification results compared to MSFragger. (**A**) Distribution of identified N-glycosylation motifs. For PaSER and MSFragger, the motif of the identified glycopeptides is shown. For Uniprot, the motif of all annotated glycosylation sites for all proteins used in the database search is shown. (**B**) Identified spectra overlap between PaSER and MSFragger. Different identification is a difference in the peptide sequence, the glycan mass, or both. * 7200 spectra, ** 3255 spectra, and *** 3614 spectra. (**C**) Overall results comparison for consistently identified (at least 8 of 10 samples) peptide moiety sequence and glycoprotein.

**Figure 5 ijms-24-07869-f005:**
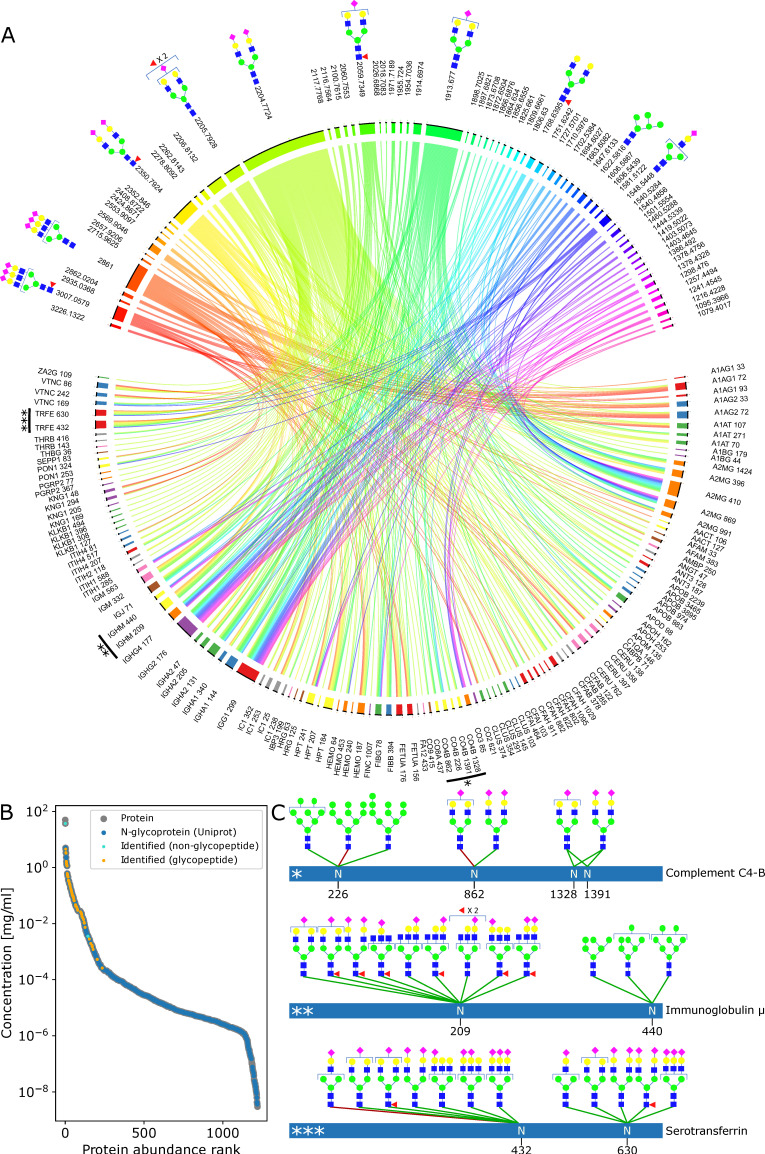
Identified glycopeptides in light of the plasma glycoproteome. (**A**) The relationship between glycosylation sites (bottom) and glycan mass (top). The glycan masses were grouped according to the GlyGen database; each group is represented by its smallest matched mass. The annotated glycans are the most probable glycan for that mass. *, **, and *** highlight the proteins in (**C**). (**B**) Glycoprotein abundance distribution (concentrations according to [22]). (**C**) Three representative glycoproteins and their identified glycans. The most probable glycan is connected to the identified glycosylation site with a green edge when it was reported in the GlyGen database or with a red edge when it was not.

**Figure 6 ijms-24-07869-f006:**
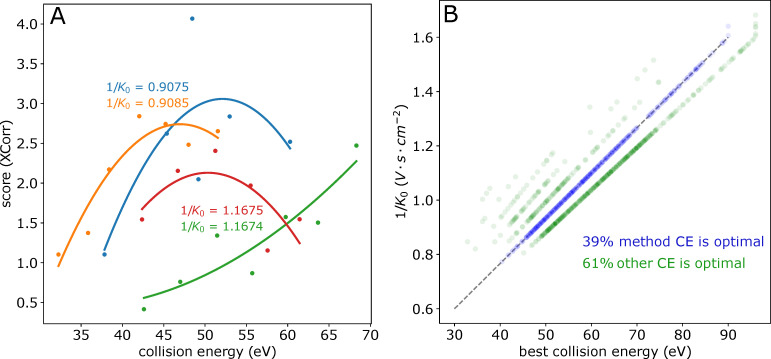
The optimal CE (collision energy) is glycopeptide-specific. (**A**) Examples for glycopeptides with very similar mobility values but different optimal CEs. The glycopeptides are as follows: Blue SVQEIQATFFYFTPNK–Hex_5_HexNAc_4_NeuAc_2_, Yellow VVLHPNYSQVDIGIK–Hex_5_HacNAc_4_NeuAc_2_, Red SLGNVNFTVSAEALESQELCGTEVPSVPEHGR–Hex_5_HexNAc_2_, and Green GLTFQQNASSMCVPDQDTAIR–Hex_5_HexNAc_4_dHex_1_NeuAc_1_. (**B**) The best-scoring glycopeptide identification out of 7 CE settings. In blue are the identifications with the optimized default CE setting, and in green are the identifications with higher or lower CE settings.

## Data Availability

We made the mass spectrometry data available at the PRIDE repository (identifier PXD040716).

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
