# Peer review of "The GlycoPaSER Prototype as a Real-Time N-Glycopeptide Identification Tool Based on the PaSER Parallel Computing Platform"

_ijms, 2023, doi:10.3390/ijms24097869_

Round 1
Reviewer 1 Report
The manuscript “The GlycoPaSER prototype as a real-time N-glycopeptide identification tool based on the PaSER parallel computing platform” presents new, real-time results dependent (RDA) approach to structure elucidation of glycoproteins to solve a central problem of glycoproteomics. It is well-known that wide range of physicochemical properties of glycopeptides lead to a wide range of optimal acquisition parameters. The authors establish GlycoPaSER prototype by extending the Parallel Search Engine in Real-time (PaSER) functionality for real-time glycopeptide identification from fragmentation spectra. Glycopeptide fragmentation spectra were decomposed into peptide- and glycan-moiety spectra using common N-glycan fragments. Each moiety was subsequently identified by a specially developed algorithm running in real-time. GlycoPaSER engine was successfully validated thus giving good results which are in fair agreement with offline MSFragger-Glyco and literature reference data. The authors claimed that GlycoPaSER prototype provided the first proof-of-concept for real-time glycopeptide identification.
Reviewer's notes. General. Figure legends are placed above the corresponding figures though legends are commonly positioned below figures.
Minor.
1. Ref. [22] is incomplete.
2. Fig. 4. Caption. Should be "peptide-moiety sequence" (“e” is missed).
3. Page 3, line 104 and below. "...peptide moiety (Y0 ion)". According to Biemann's nomenclature of fragment ions in mass spectra of peptides, small letters are used. So, it should be designated as y0. Capital letters are applicable in Domon-Costello nomenclature of cleavages in oligosaccharide ions (reference [24] in this manuscript). This difference cannot be neglected otherwise, a reader may be confused.
4. The last page in "Supplementary Materials" is void. Something is missed?

Reviewer 2 Report
GlycoPaSER shows promise as a means for evaluating glycopeptide MS/MS in real time, and as a proof-of-concept performs admirably on the test data. The authors note some areas of refinement such as incorporating glycan spectral patterns instead of just glycan mass, which would help close the gap between this test and the MSFragger results. But for a paper ostensibly focused on presenting a new method, it seems very sparse on the actual methodology. The text includes lines such as "We evaluated several parameters to find the correct fragmentation pattern and their optimal values based on the plasma data. [186-187]" but no details of that evaluation were given. Comparisons with existing software are supported with graphs and numbers, but the approach is not given the same attention.
The results section shifts from checking for oxonium ion patterns to matching precursor patterns with no explanation of what this precursor pattern matching step entailed. Once these patterns are matched, the top matches are used to make pseudo spectra which are then fed to PaSER as a sort of competitive search step. How were these modified spectra made and how do they compare to the unmodified spectrum? Given that most of the assignments from MSFragger that were not reproduced by GlycoPaSER were due to a peptide mismatch, more details seem vital here to evaluating the efficacy of this search step. The peptide fragments are much lower intensity than the oxonium and peptide+Y ions—which is evident when looking at Figure S2 but obfuscated when looking at the idealized spectra presented in Figure 1b of the main text—which is something that also warrants discussion when choosing to evaluate modified spectra. Finally, I would appreciate better organization of this section—if the heading is about oxonium ion filtering, then I do not expect to be reading about peptide assignment.
Some of the figures feel superfluous or overdesigned. Figure 3 takes up half a page to essentially convey four numbers describing the performance speed. The circle itself is decoration at best, and at worst a confusing way to illustrate the processes, especially compared with the earlier process diagram of Figure 1.
The chord diagram in Figure 5A is invoked as a sort of sanity check—are the most frequent glycan configurations what we would expect from plasma samples—but this could be much more cleanly conveyed in a table or even simply reported in the text itself. One might attempt to follow an individual line if they were interested in a particular pairing, but this is much better illustrated by the representative glycoproteins you've already extracted for 5C.
The utility of a tool like GlucoPaSER is well-established, and it is clear that this prototype puts us much closer to real-time evaluation. But it is difficult to comment on the approach as the text is overly results-focused.
